# Closing the Gap Between Text and Speech Understanding in LLMs

**Santiago Cuervo**[†*]**, Skyler Seto**[‡]**, Maureen de Seyssel**[‡]**, Richard He Bai**[‡]**, Zijin Gu**[‡]**,
Tatiana Likhomanenko**[‡]**, Navdeep Jaitly**[‡]**, Zakaria Aldeneh**[‡]

[†]Université de Toulon, Aix Marseille Université, CNRS, LIS
[‡]Apple

## Abstract

Large Language Models (LLMs) can be adapted to extend their text capabilities to speech inputs. However, these speech-adapted LLMs consistently underperform their text-based counterparts—and even cascaded pipelines—on language understanding tasks. We term this shortfall the *text–speech understanding gap*: the performance drop observed when a speech-adapted LLM processes spoken inputs relative to when the original text-based LLM processes the equivalent text. Recent approaches to narrowing this gap either rely on large-scale speech synthesis of text corpora, which is costly and heavily dependent on synthetic data, or on large-scale proprietary speech datasets, which are not reproducible. As a result, there remains a need for more data-efficient alternatives for closing the text-speech understanding gap. In this work, we analyze the gap as driven by two factors: (i) forgetting of text capabilities during adaptation, and (ii) cross-modal misalignment between speech and text. Based on this analysis, we introduce SALAD—**S**ample-efficient **A**lignment with **L**earning through **A**ctive selection and cross-modal **D**istillation— which combines cross-modal distillation with targeted synthetic data to improve alignment while mitigating forgetting. Applied to 3B and 7B LLMs, SALAD achieves competitive performance with a strong open-weight model across broad-domain benchmarks in knowledge, language understanding, and reasoning, while training on over an order of magnitude less speech data from public corpora.

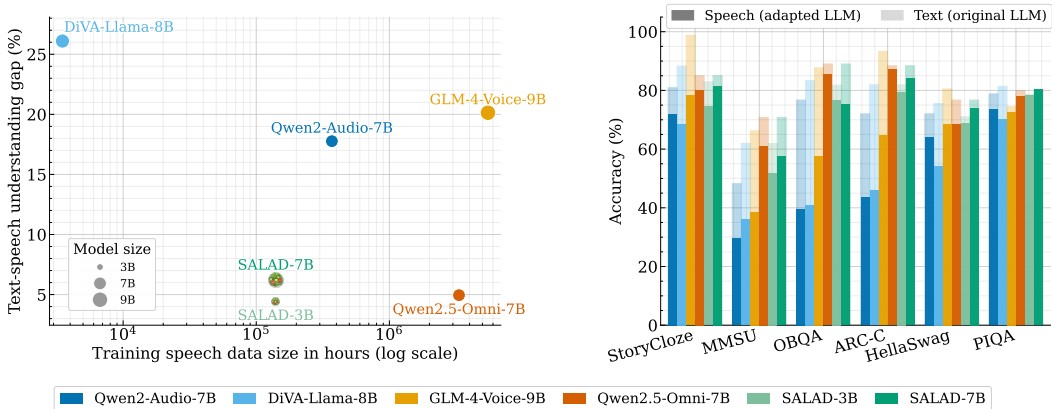

Figure 1: SALAD reduces the text-speech understanding gap on general reasoning and language understanding benchmarks while requiring over an order of magnitude less training data than competing speech-adapted LLMs. *Left:* The gap between the performance of the base LLM given text inputs and the performance of the speech-adapted LLM given speech inputs as a function of training dataset size[1]. *Right:* The performance of the base LLM given text inputs and the performance of the speech-adapted LLM given speech inputs across several language understanding tasks.

---

[*]Work done during an internship at Apple.

# 1 INTRODUCTION

Large language models (LLMs) have demonstrated impressive capabilities in general knowledge and reasoning, often surpassing specialized systems across a wide range of tasks. This success has motivated efforts to extend LLMs to the speech domain, opening new possibilities for spoken natural interaction. A straightforward approach to extend LLMs to the speech domain is the cascaded pipeline, where automatic speech recognition (ASR) models map speech to text and the LLM is applied to the transcribed text. While effective in preserving text capabilities, cascaded pipelines largely remove speaker and paralinguistic cues essential for natural spoken interaction (Maimon et al., 2025b). To address this limitation, recent work has explored end-to-end approaches, adapting text-based LLMs to directly process speech inputs (Tang et al., 2024; Chu et al., 2024; Xie & Wu, 2024; Fang et al., 2024; Nguyen et al., 2024; Défossez et al., 2024). Despite their promise, speech-adapted LLMs struggle with the core requirement of language understanding, consistently underperforming text-based LLMs and even cascaded systems on language understanding tasks (Cuervo & Marxer, 2024; Chen et al., 2024; Cui et al., 2025). We refer to this shortfall as the *text–speech understanding gap*: the performance drop when a speech-adapted LLM performs a language understanding task in the speech domain compared to the original LLM performing the same task in the text domain. Closing this gap is a crucial step toward building AI systems capable of truly natural spoken interaction.

Prior work has proposed several strategies to reduce this gap. A common direction is cross-modal alignment, achieved either by optimizing the fusion between speech encoders and text LLMs to promote modality-agnostic representations (Tang et al., 2024; Deng et al., 2025; Held et al., 2025; Tseng et al., 2025) or by explicitly training for consistent outputs across modalities given equivalent inputs (Fathullah et al., 2024; Wang et al., 2024; Held et al., 2025). Another direction is data-driven methods, which synthesize large-scale speech from text corpora to narrow the distribution gap between speech and text training domains (Zeng et al., 2025). While these methods show performance improvements, they focused on narrow-domain benchmarks and several did not evaluate performance relative to the original text-based LLMs. When evaluated on broader benchmarks, these approaches showed substantial drops compared to their text-based LLM backbones (Chen et al., 2024). Despite these efforts, the text-speech understanding gap persists. More recent methods (Xu et al., 2025; KimiTeam et al., 2025) demonstrated notable progress, but remain irreproducible due to missing training details and their reliance on massive proprietary speech datasets spanning millions of hours—equivalent to over 100B text tokens[2], i.e., a budget comparable to full pretraining budgets in the text domain. Given the scarcity of publicly available speech and parallel speech–text data relative to text data, more sample-efficient methods are needed.

In this work, we seek to understand the text–speech understanding gap in greater depth to better guide the design of efficient remedies. Bridging the text–speech understanding gap requires a speech-adapted LLM to retain the knowledge of its text-based counterpart (i.e., avoid forgetting) and produce consistent outputs for equivalent speech and text inputs (i.e., avoid cross-modal misalignment). Forgetting refers to the loss of pretrained text capabilities during adaptation to speech, a well-documented effect of domain shift between pretraining and fine-tuning in LLMs (Béthune et al., 2025). Cross-modal misalignment is observed when semantically equivalent speech and text inputs give divergent outputs. We quantify forgetting and cross-modal misalignment, and study these measures under different training objectives and data regimes. The gained insights lead us to propose SALAD: **S**ample-efficient **A**lignment with **L**earning through **A**ctive selection and cross-modal **D**istillation, a sample-efficient method to address the performance gap. Our main contributions are:

- We quantify forgetting and cross-modal misalignment as the statistical distances between the outputs of a speech-adapted LLM and its text-based LLM backbone on matched text–speech inputs from a broad-domain pretraining corpus, and show that these measures are highly predictive of the text–speech understanding gap on broad-domain benchmarks.
- We find that training on narrow-domain speech corpora with standard objectives used in prior work worsens both forgetting and broad-domain misalignment as the amount of training data increases. In contrast, using a cross-modal knowledge distillation objective—where the text-

---

[1]For Qwen2.5-Omni, we approximate the dataset size in hours based on the reported number of pretraining audio tokens. However, the proportion of this data that is actual speech is not reported.

[2]Estimated from the average duration of text tokens (∼320 ms) in our data.

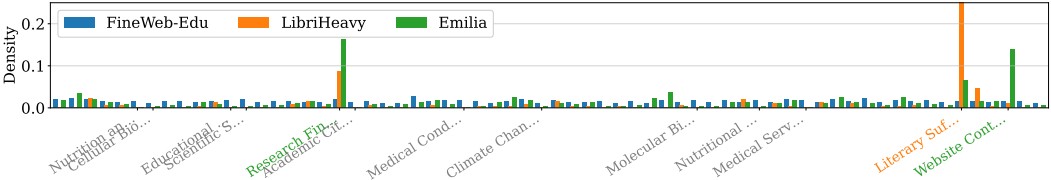

Figure 2: Distribution of samples across 64 automatically annotated domains (x-axis). Text corpora (FineWeb-Edu; Penedo et al., 2024) cover a wide range of domains, whereas speech datasets (LibriHeavy; Kang et al., 2024 and Emilia; He et al., 2024) are concentrated in only a few. The ten least represented domains in speech are labeled in grey.

based LLM backbone serves as the teacher—improves alignment and mitigates forgetting, even when trained on narrow-domain data.

- We show that cross-modal distillation alone leaves residual misalignment when trained only on narrow-domain data. To further address this misalignment, we introduce an active learning algorithm that targets domain mismatches between natural speech and text corpora.

- We propose a method that builds on insights from our analyses, apply it to 3B and 7B LLMs, and benchmark them against recent speech-adapted LLMs in the 3B–9B range on spoken versions of six broad-domain knowledge and reasoning benchmarks. Our models outperform most baselines in spoken language understanding and perform competitively with the strongest, while training on over an order of magnitude less data.

## 2 PRELIMINARIES

Recent end-to-end methods for speech-adapted LLMs typically generate text as an intermediate representation conditioned on speech inputs, and then generate speech conditioned on that text (Xie & Wu, 2024; Défossez et al., 2024; Fang et al., 2024; Xu et al., 2025; KimiTeam et al., 2025). In this work, we focus on generating intermediate text conditioned on speech input as our primary task since it directly reflects language capability, leaving the task of generating speech for future work. Specifically, we aim to build a speech-adapted language model $P_\theta$, parameterized by weights $\theta$, that defines a probability distribution over the next text token given a multimodal context. Let $\boldsymbol{c}$ denote such a context, which may contain subsequences of text tokens $\boldsymbol{w}$ and/or speech representations $\boldsymbol{a}$. For each position $i$, the model predicts the distribution over the next text token $w_{i+1}$ conditioned on $\boldsymbol{c}_{\leq i}$: $P_\theta(w_{i+1} \mid \boldsymbol{c}_{\leq i})$.

We seek to train $P_\theta$ so that its predictions match the distribution of $(\boldsymbol{w}, \boldsymbol{c})$ pairs drawn from the natural language distribution $\mathcal{Q}$. Most prior works approximate $\mathcal{Q}$ by minimizing the negative log-likelihood on a speech dataset $\mathbb{D}$ (Zhang et al., 2023; Nguyen et al., 2024; Défossez et al., 2024; Chu et al., 2024; Zeng et al., 2025):

$$- \sum_{(\boldsymbol{w}, \boldsymbol{c}) \in \mathbb{D}} \log P_\theta(w_{i+1} \mid \boldsymbol{c}_{\leq i}). \tag{1}$$

For this approach to succeed, the dataset $\mathbb{D}$ must be a sufficiently representative sample of $\mathcal{Q}$. However, available speech datasets are narrow in scope and fail to capture the full general language distribution. This limitation is evident in the domain distribution shown in Figure 2.

Transfer learning is used to alleviate this limitation, with $P_\theta$ initialized from a pretrained text-only language model $Q_\phi$—which better approximates $\mathcal{Q}$—in the hope of enabling general knowledge transfer even when fine-tuned on narrow speech data. In practice, however, models trained this way under-perform on spoken language understanding tasks relative to $Q_\phi$ (Chen et al., 2024; Cui et al., 2025). One contributor to this under-performance is *cross-modal misalignment*, i.e., inconsistent predictions across modalities, which we formalize as

$$M = \sum_{(\boldsymbol{w}, \boldsymbol{c}) \sim \mathcal{Q}} D_{\mathrm{KL}}\big(P_\theta(w_{i+1} \mid \boldsymbol{w}_{\leq i}) \,\big\|\, P_\theta(w_{i+1} \mid \boldsymbol{c}_{\leq i})\big), \tag{2}$$

where $D_{\text{KL}}(\cdot|\cdot)$ denotes the Kullback–Leibler divergence between two distributions, $\boldsymbol{w}$ denotes a text sequence, $\boldsymbol{c}$ denotes a semantically equivalent multimodal context sequence, and $w_{i+1}$ denotes the next text token. This quantity measures how differently the model predicts the next token when conditioned on text versus equivalent multimodal context.

Another contributor to this under-performance on spoken language understanding tasks relative to $Q_\phi$ is *forgetting*, which measures the loss of the original text behavior:

$$F = \sum_{\boldsymbol{w} \sim \mathcal{Q}} D_{\text{KL}}\big(Q_\phi(w_{i+1} \mid \boldsymbol{w}_{\leq i}) \,\big\|\, P_\theta(w_{i+1} \mid \boldsymbol{w}_{\leq i})\big). \tag{3}$$

This quantity measures how much the speech-adapted model $P_\theta$ diverges from the text-based LLM $Q_\phi$ on text inputs, indicating loss of text knowledge and reduced ability to transfer capabilities to the speech domain.

## 3 ANALYZING THE TEXT-SPEECH GAP

In this section, we study how cross-modal misalignment (Equation 2) and forgetting (Equation 3) in speech-adapted LLMs affect downstream language understanding performance, and how different design decisions impact these two metrics. Specifically, we train multiple models while varying training objectives, datasets, and training budgets. Then, for each trained model, we measure cross-modal alignment, forgetting, and the performance on broad-domain language understanding benchmarks.

### 3.1 OBJECTIVE

We train our models in a pretraining setup, modeling general sequences without the data templates used in instruction-tuned models. This choice reflects our focus on broad-domain alignment, avoiding restriction to dialogue data and acknowledging the scarcity of speech instruction data. Our data therefore consist of multimodal sequences $\boldsymbol{c}$ composed of interleaved speech $\boldsymbol{a}$ and text $\boldsymbol{w}$ inputs, as in Nguyen et al. (2024). For our training objective, we introduce a variable $\alpha \in [0, 1]$ that interpolates between a standard maximum likelihood objective, and a cross-modal distillation objective, similar to that used by Wang et al. (2024); Held et al. (2025):

$$\mathcal{L}(\mathbb{D}, \theta) = \alpha\, \mathcal{L}_{\text{DIST}}(\mathbb{D}, \theta) + (1 - \alpha)\, \mathcal{L}_{\text{NLL}}(\mathbb{D}, \theta), \tag{4}$$

where

$$\mathcal{L}_{\text{DIST}}(\mathbb{D}, \theta) = \sum_{(\boldsymbol{w}, \boldsymbol{c}) \in \mathbb{D}} \sum_i \mathbf{1}_{\{\text{text at i+1}\}}\, D_{\text{KL}}\big(Q_\phi(w_{i+1} \mid \boldsymbol{w}_{\leq i}) \,\big\|\, P_\theta(w_{i+1} \mid \boldsymbol{c}_{\leq i})\big), \tag{5}$$

$$\mathcal{L}_{\text{NLL}}(\mathbb{D}, \theta) = -\sum_{(\boldsymbol{w}, \boldsymbol{c}) \in \mathbb{D}} \sum_i \mathbf{1}_{\{\text{text at i+1}\}}\, \log P_\theta(w_{i+1} \mid \boldsymbol{c}_{\leq i}). \tag{6}$$

Here, $\mathbf{1}_{\{\text{text at i+1}\}}$ is an indicator function equal to 1 if the $(i+1)$-th element in the interleaved context is a text token, and 0 otherwise.

### 3.2 DATA

We consider the two natural English speech corpora LibriHeavy (Kang et al., 2024) (read speech) and the YODAS-EN subset of the Emilia dataset (He et al., 2024) (conversational), both among the largest and most semantically diverse publicly available speech datasets. However, as shown in Figure 2, they still lack domain coverage relative to text pretraining corpora. Since forgetting is driven by domain shift, we also synthesize a spoken version of a 10B-text-token subset of FineWeb-Edu (Penedo et al., 2024)—a high-quality broad-domain text pretraining corpus—to study the impact of aligning text and speech training domains, following the approach of Zeng et al. (2025).

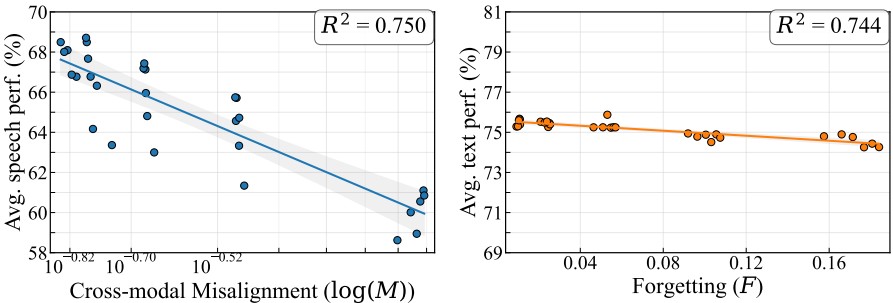

Figure 3: Relationship between speech performance and cross-modal misalignment (Equation 2) (left); relation between text performance and forgetting (Equation 3) (right).

We use the Kokoro-TTS model[3] with the `af-heart` voice, which provides the highest quality generations[4], to synthesize the data.

To produce interleaved speech–text sequences, we segment each utterance into subsequences and interleave spans of random length at runtime: 10–30 words for text segments and 5–15 words for speech segments. For LibriHeavy and Emilia, word-level timestamps of the corresponding transcriptions are obtained using the forced aligner from Pratap et al. (2024). For synthesized speech, we use the built-in functionality of Kokoro TTS to get word-level timestamps.

## 3.3 BENCHMARKS

We evaluate on broad-domain benchmarks of general knowledge, reasoning, and language understanding commonly used for LLMs, considering both their text and spoken versions: StoryCloze (Hassid et al., 2023), MMSU and OpenBookQA (OBQA) from VoiceBench (Chen et al., 2024), HellaSwag (Zellers et al., 2019), ARC-Challenge (Clark et al., 2018), and PIQA (Bisk et al., 2020). We adopt a few-shot prompting approach for evaluation across all tasks, with accuracy as the metric. For further details on the benchmarks, see Appendix A.5.

## 3.4 ARCHITECTURE

We follow the standard design of speech-adapted LLMs (Arora et al., 2025), consisting of a speech encoder that extracts representations from waveforms, an adapter that maps them into the input space of the language model, and the language model itself. For these experiments we initialize $P_\theta$ from the text LLM Qwen2.5-3B (Qwen et al., 2025) base model, and use the same text LLM as teacher $Q_\phi$ in Equation 5, allowing us to measure how much original text capability is maintained when processing speech and how much is lost during speech training.

While prior work has paid great attention to speech encoder and adapter design—typically to promote cross-modal alignment by making speech representations more text-like (Tang et al., 2024; Deng et al., 2025; Held et al., 2025; Tseng et al., 2025)—we adopt a simple architecture: the lightweight Mimi speech tokenizer (Défossez et al., 2024) as encoder and a 122M-parameter stack of transformer decoder layers as adapter. We make this choice because current representation alignment methods rely on large non-causal encoders and complex modules unsuited for low-latency streaming, which is essential for downstream conversational speech-adapted LLMs (Défossez et al., 2024). Accordingly, we use causal, streaming-friendly models with low-level, non–text-like representations as a "worst-case" input alignment scenario, expecting our findings to generalize to more aligned text-speech representations while being directly applicable to low-latency architectures. The encoder remains frozen during training, while the adapter and language model are optimized. For further details on the model architecture, see Appendix A.1.

---

[3] https://github.com/hexgrad/kokoro-82M
[4] https://huggingface.co/hexgrad/Kokoro-82M/blob/main/VOICES.md

## 3.5 RESULTS

We train our models by selecting $\alpha$ from $\{0, 0.25, 0.5, 0.75, 1\}$, selecting training data $\mathbb{D}$ from $\{\text{Emilia+LibriHeavy, FineWeb-Edu}\}$, and adjusting training budget $D$ within the range (2B–12B tokens). Each model is trained using the hyperparameters described in Appendix A.2. For each trained model, we measure cross-modal misalignment (Equation 2) and forgetting (Equation 3) on a test subset of our text–speech version of FineWeb-Edu, and calculate the average accuracy on the broad-domain benchmarks. We present the results below.

**How do cross-modal misalignment and forgetting relate to broad-domain performance?** Figure 3 shows scatter plots with ordinary least squares fits for models trained on narrow-domain speech data (LibriHeavy + Emilia): (i) average speech performance (%) vs. $\log(\text{misalignment})$, and (ii) average text performance (%) vs. forgetting. The solid line indicates the fitted regression; the shaded band is the $95\%$ confidence interval for the mean prediction. Each panel also reports the Leave-One-Out Cross-

Table 1: Univariate and partial $R^2$ explaining the variance in speech and text performance, attributed to forgetting and misalignment.

| Predictor | Speech perf. $R^2$ | Text perf. $R^2$ |
|---|---|---|
| Forgetting | 0.455 | **0.744** |
| Misalignment | **0.750** | 0.614 |
| Misalignment \| Forgetting | **0.563** | -0.001 |
| Forgetting \| Misalignment | 0.049 | **0.323** |

Validation (LOOCV) $R^2$ of the univariate fit. The results show that speech performance declines with increasing misalignment (LOOCV $R^2 = 0.75$), and text performance declines with increasing forgetting (LOOCV $R^2 = 0.74$). Table 1 reports both univariate and partial $R^2$ (i.e., variance explained when adding the other factor). Misalignment uniquely explains a large share of speech variance given forgetting (partial $R^2 \approx 0.56$), while forgetting uniquely explains text variance given misalignment (partial $R^2 \approx 0.32$). These patterns hold when controlling for $\alpha$ and training budget, with unique cross-validated $R^2$ of $\sim 0.34$ for speech (misalignment) and $\sim 0.23$ for text (forgetting). For more details on the least squares analysis, see Appendix A.6.

**How does the training objective interact with cross-modal alignment and forgetting?** Figure 4 reports scaling curves for cross-modal misalignment, forgetting, and average speech performance. Training with NLL (i.e., $\alpha = 0$ in Equation 4) on narrow-domain speech data (LibriHeavy + Emilia) leads to increasing cross-modal misalignment with scale. Given the strong relation between misalignment and speech performance shown in Figure 3, NLL training on narrow-domain data also yields the weakest results. NLL training leads to greater forgetting of the pretrained text behavior compared to models trained with nonzero $\alpha$ values. This greater forgetting, however, has limited impact on the average text performance (see Appendix A.7).

Higher values of $\alpha$ yield better alignment, and for $\alpha > 0$, training on narrow-domain data generalizes to reduced broad-domain misalignment with scale. Misalignment is well described by a typical log-linear neural scaling law (Kaplan et al., 2020; Hoffmann et al., 2022). We fit scaling laws of misalignment with respect to the training budget $D$ of the form $M = E + B\,D^{-\beta}$, where $E$ denotes the irreducible misalignment and $B$ and $\beta$ capture scaling efficiency. The fitted laws are reported in Table 2, along with LOOCV $R^2$ and the estimated number of tokens required to reach within 5% of $E$. The irreducible misalignment $E$ decreases with $\alpha$, and for all $0 < \alpha < 1$, misalignment saturates early in training. Distillation is therefore the most scalable approach

Table 2: Scaling laws of misalignment with training tokens ($D$) for runs with $\alpha > 0$: $M = E + B\,D^{-\beta}$. Reported are the LOOCV $R^2$ and $D$ needed to be within 5% of $E$.

| $\alpha$ | $E$ | $B$ | $\beta$ | $R^2_{\text{LOOCV}}$ | Tokens@5%E |
|---|---|---|---|---|---|
| *LibriHeavy + Emilia* | | | | | |
| 0.25 | 0.32 | 45.1 | 0.46 | 0.81 | 4.94B |
| 0.50 | 0.21 | 3.2e10 | 1.34 | 0.75 | 2.07B |
| 0.75 | 0.16 | 1.2e8 | 1.04 | 0.85 | 6.00B |
| 1.00 | 0.13 | 3806 | 0.54 | 0.91 | 47.58B |
| *FineWeb-Edu* | | | | | |
| 1.00 | 0.04 | 5.4 | 0.56 | 0.96 | 268.4B |

with respect to misalignment. Although forgetting increases slightly with scale regardless of $\alpha$, this effect has limited impact on performance.

**Does a better data domain match between speech and text training alleviate the issues of NLL training?** Figure 4 shows that training on broad-domain data (FineWeb-Edu) with $\alpha = 0$ reduces misalignment relative to narrow-domain data, although misalignment still grows with scale. This indicates that domain matching alone does not resolve cross-modal misalignment. Counterintuitively,

forgetting is slightly worse than with narrow-domain training, but the difference is small. Speech performance is not tied to misalignment, with the model outperforming others despite higher misalignment. However, while $\alpha > 0$ yields consistent improvements with scale, NLL training on broad-domain data shows no meaningful scaling gains.

**Does a better data domain match between speech and text training yield gains for cross-modal distillation?** Figure 4 presents the results with $\alpha = 1$ and broad-domain (FineWeb-Edu-train) training data. In this setting, misalignment is essentially the quantity being optimized—distillation directly targets cross-modal consistency—and, since the evaluation distribution (FineWeb-Edu-test) matches the training distribution, the metric aligns with the objective. Accordingly, domain-matched distillation yields the lowest misalignment and the strongest speech-understanding performance.

**Takeaways.** Our analyses highlight two central insights: (i) cross-modal knowledge distillation objective is more effective than maximum likelihood for mitigating misalignment and forgetting (Figure 4, Table 2), and (ii) better matching the domain of speech training data to that of text pretraining yields further gains when combined with cross-modal distillation. We use these insights to design a learning strategy to address the text-speech understanding gap in the next section.

## 4 CLOSING THE TEXT-SPEECH GAP

Natural speech corpora are narrower in terms of domain coverage compared to text pretraining corpora (Figure 2). Large-scale synthetic speech (i.e., same size as text pretraining corpora), while useful for improving the domain coverage, is both costly and lacking in the paralinguistic richness essential for natural spoken interaction (Debnath et al., 2024; Minixhofer et al., 2025). It is therefore desirable to reduce reliance on large-scale synthetic speech data. To this end, we propose SALAD: **S**ample-efficient **A**lignment with **L**earning through **A**ctive selection and cross-modal **D**istillation. SALAD is designed directly around our two key insights: distillation ensures robust alignment and mitigates forgetting, while active selection enables closer domain matching through a minimal, model-guided infusion of synthetic speech data rather than costly large-scale synthesis.

### 4.1 METHOD

We structure SALAD as a two-stage process. **Stage I (Distillation on Natural Speech)** trains $P_\theta$ on natural speech by minimizing $\mathcal{L}_{\text{DIST}}$ between $P_\theta$ and $Q_\phi$ on $\mathbb{D}_{\text{speech}}$ (Equation 5), leveraging the strong scaling behavior of distillation until alignment plateaus at the irreducible misalignment $E$, which, as shown in Table 2, occurs within a practical training budget. **Stage II (Active Selection for Domain Expansion)** then addresses the residual misalignment by introducing a small but strategically chosen amount of synthetic speech through *active selection*, guided by the model's own misalignment signals. This targeted augmentation complements Stage I by expanding domain

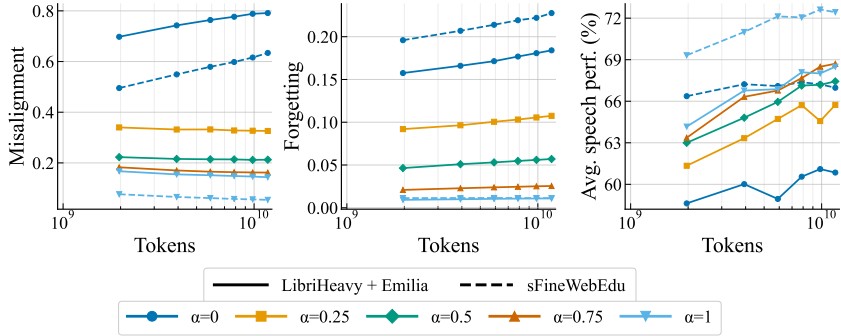

Figure 4: Impact of the training objective (controlled by $\alpha$ in Equation 4), the number of training tokens (shown on the $x$-axis), and the dataset choice on cross-modal misalignment (Equation 2), forgetting (Equation 3), and average speech performance.

coverage while keeping reliance on synthetic data minimal, consistent with our emphasis on sample efficiency and the use of natural speech.

First, we aim to develop a sampling strategy for choosing which text samples to synthesize. We draw inspiration from CRISP (Grangier et al., 2025), adopting a clustered importance-sampling strategy that derives a target-domain dataset from a broad-domain corpus by reweighting clusters. Concretely, let $\mathbb{D}_{\text{web}}$ be a large broad-domain text corpus. We partition it into $K$ clusters $K(c)_{c=1}^{K}$ using sentence embeddings and $k$-means, and define

$$P_{\text{web}}(c) \;=\; \frac{|\mathbb{D}_{\text{web}}^{(c)}|}{|\mathbb{D}_{\text{web}}|}, \qquad \mathbb{D}_{\text{web}}^{(c)} \;=\; \{\, \boldsymbol{w} \in \mathbb{D}_{\text{web}} : \boldsymbol{w} \in K(c) \,\}. \tag{7}$$

We compute per-cluster importance weights as $w(c) = \frac{P_{\text{target}}(c)}{P_{\text{web}}(c)}$. In CRISP, $P_{\text{target}}(c)$ comes from the distribution of a small target-domain dataset across the clusters. In our case, no such dataset exists, so we let the model itself define $P_{\text{target}}$. Specifically, we treat the divergence between $P_\theta$ and $Q_\phi$ within each cluster as a proxy for how much that cluster belongs to the "missing" domain. We define

$$P_{\text{target}}(c) \;=\; \frac{P_{\text{web}}(c)\, f_\gamma(M(c))}{\sum_{c'=1}^{K} P_{\text{web}}(c')\, f_\gamma(M(c'))}, \qquad f_\gamma(m) \;=\; m^\gamma, \tag{8}$$

where $M(c) \;=\; \mathcal{L}_{\text{DIST}}(\mathbb{D}_{\text{probe}}^{(c)})$ is the misalignment at cluster $c$, measured on a small probe set $\mathbb{D}_{\text{probe}}^{(c)} \subset \mathbb{D}_{\text{web}}^{(c)}$ for which we pre-synthesize speech. Clusters with higher misalignment—where $P_\theta$ diverges most from $Q_\phi$—are therefore upweighted. The resulting importance weight is

$$w_\gamma(c) \;=\; \frac{P_{\text{target}}(c)}{P_{\text{web}}(c)} \;\propto\; f_\gamma(M(c)). \tag{9}$$

The parameter $\gamma \geq 0$ controls the trade-off: $\gamma = 0$ reduces to sampling purely from $P_{\text{web}}$, while larger values focus more heavily on clusters with greater misalignment.

We sample clusters in proportion to their importance rather than reweighting per-example losses, thereby avoiding the need to synthesize and train on the entire $\mathbb{D}_{\text{web}}$. Given a fixed synthesis budget, we repeatedly draw a cluster $c$ according to $P_{\text{target}}$, select a text sequence $\boldsymbol{w}$ uniformly from $\mathbb{D}_{\text{web}}^{(c)}$, synthesize its speech $\boldsymbol{a}$, and form an interleaved context $\boldsymbol{c}$. Each sample is added to the active dataset $\mathbb{D}_{\text{active}}$ until the budget is exhausted, after which $\mathbb{D}_{\text{active}}$ is used to continue training $P_\theta$. To prevent forgetting of Stage I training, we combine $\mathbb{D}_{\text{active}}$ with $\mathbb{D}_{\text{speech}}$ and minimize $\mathcal{L}_{\text{DIST}}(\mathbb{D}_{\text{speech}} \cup \mathbb{D}_{\text{active}}, \theta)$.

## 4.2 SETUP

We apply our method to the Qwen2.5 3B and 7B base LLMs (Qwen et al., 2025), yielding the SALAD-3B and SALAD-7B models. We follow the experimental setup of Section 3 for architecture, data, and evaluation tasks. We use Emilia and LibriHeavy ($141,612$ hours) as our $\mathbb{D}_{\text{speech}}$, and use a 10B-token FineWeb-Edu subset as our $\mathbb{D}_{\text{web}}$. For Stage II, we train a clustering model on $\mathbb{D}_{\text{web}}$ using balanced $k$-means with $K = 128$ over `BAAI/bge-large-en-v1.5` embeddings[5], with a synthesis budget of $1\%$ of $\mathbb{D}_{\text{speech}}$ and $\gamma = 5$ (Equation 8). We train SALAD models for $24$B tokens during Stage I and additional 1.9B tokens during Stage II. Further training details are provided in Appendix A.3, and additional ablations are reported in Appendix A.8.

We benchmark SALAD models against the following speech-adapted LLMs from the literature: Qwen2-Audio (Chu et al., 2024), DiVA (Held et al., 2025), GLM-4-Voice (Zeng et al., 2025), and Qwen2.5-Omni (Xu et al., 2025). We also evaluate against a cascaded pipeline that pairs Whisper-Large-v3 ASR with the Qwen2.5 LLMs used as the backbones of Qwen2.5-Omni and our SALAD models. The cascade pipeline serves as our topline reference for spoken language understanding. For each model, we report the per-task accuracy as well as the text–speech gap, defined as the difference between the performance of a speech-adapted LLM given speech input and the performance of the original text-based LLM given the corresponding text input.

Table 3: SALAD outperforms most baselines and is competitive with the strongest speech-adapted LLMs and cascaded pipelines. "Acc." denotes the accuracy of the speech-adapted LLM given speech input; "Gap" denotes the difference between the accuracy of the text-based LLM given text input and the "Acc." column. Gap cells are color-coded from best (green) to worst (red) for each task, where lower is better.

| | StoryCloze | | MMSU | | OBQA | | HellaSwag | | ARC-C | | PIQA | | Avg. | |
|---|---|---|---|---|---|---|---|---|---|---|---|---|---|---|
| | Acc. | Gap | Acc. | Gap | Acc. | Gap | Acc. | Gap | Acc. | Gap | Acc. | Gap | Acc. | Gap |
| Random | 50.0 | - | 25.0 | - | 25.0 | - | 25.0 | - | 25.0 | - | 50.0 | - | 33.3 | - |
| *Cascaded **toplines**: ASR (Whisper-v3-Large) + LLM* | | | | | | | | | | | | | | |
| ASR + Qwen2.5-3B | 82.7 | 0.2 | 58.1 | 3.8 | 76.9 | 4.9 | 69.1 | 1.9 | 77.7 | 4.2 | 78.5 | 0.1 | 73.8 | 2.5 |
| ASR + Qwen2.5-7B | 84.2 | 0.8 | 67.1 | 3.7 | 84.0 | 5.0 | 74.7 | 2.0 | 86.5 | 1.9 | 79.9 | 0.0 | 79.4 | 2.2 |
| *End-to-end systems* | | | | | | | | | | | | | | |
| Qwen2-Audio-7B | 71.9 | 9.0 | 29.5 | 18.7 | 39.6 | 37.1 | 64.1 | 7.9 | 43.5 | 28.5 | 73.4 | 5.4 | 53.7 | 17.8 |
| DiVA-Llama3.1-8B | 68.6 | 19.7 | 36.1 | 25.8 | 40.9 | 42.4 | 54.2 | 21.3 | 45.9 | 36.0 | 70.0 | 11.4 | 52.6 | 26.1 |
| GLM-4-Voice-9B | 78.2 | 20.6 | 38.6 | 27.6 | 57.6 | 30.1 | 68.6 | 11.9 | 64.6 | 28.7 | 72.6 | 1.9 | 63.4 | 20.1 |
| Qwen2.5-Omni-7B | 80.1 | 4.9 | **61.0** | -9.8 | **85.5** | **3.5** | 68.4 | 8.3 | **87.1** | **1.3** | 78.0 | 1.9 | **76.7** | 5.0 |
| SALAD-3B | | | | | | | | | | | | | | |
|   Stage I | 75.5 | 7.4 | 47.3 | 14.6 | 65.5 | 16.3 | 68.8 | 2.2 | 75.6 | 6.2 | 78.3 | 0.3 | 68.5 | 8.0 |
|   Stage II | 75.8 | 7.1 | 52.5 | **9.4** | 76.7 | 5.1 | 68.7 | **2.3** | 79.9 | 1.9 | 78.1 | 0.5 | 72.0 | **4.6** |
| SALAD-7B | | | | | | | | | | | | | | |
|   Stage I | **81.5** | **3.5** | 55.3 | 15.5 | 69.7 | 19.3 | **74.2** | 2.5 | 82.3 | 6.1 | **80.3** | **0.4** | 73.9 | 7.9 |
|   Stage II | **81.5** | **3.5** | 57.5 | 13.3 | 75.1 | 13.9 | 74.0 | 2.7 | 84.0 | 4.4 | **80.3** | **0.4** | 75.4 | 6.2 |

## 4.3 RESULTS

Table 3 summarizes the performance of our approach compared to existing baselines and the cascaded pipeline topline. Overall, SALAD achieves performance competitive with the strongest model, Qwen2.5-Omni, while using over an order of magnitude less speech data (Figure 1). In particular, SALAD-3B outperforms all larger end-to-end baselines except Qwen2.5-Omni, showing that strong text–speech alignment can be achieved with much smaller models when combined with our training strategy. Relative to cascaded toplines, the strongest SALAD models are competitive, underperforming them only slightly while retaining the advantages of end-to-end modeling.

We next ask whether targeting the misaligned domains is responsible for the performance gains we observe. Table 4 shows the effect of Stage II training when using our active data selection strategy compared to uniform sampling. Active data selection shows greater gains in MMSU, OpenBookQA, and ARC-C. These tasks involve scientific questions and more technical terminology than the others, making them

Table 4: Performance (Accuracy, %) of the SALAD-3B model after Stage II training with active selection vs. random (uniform) selection.

| | SC | MMSU | OBQA | HellaSwag | ARC-C | PIQA |
|---|---|---|---|---|---|---|
| Uniform | 75.0 | 49.5 | 71.9 | 68.6 | 78.9 | 78.1 |
| Active Sel. | **75.8** | **52.5** | **76.7** | **68.7** | **79.9** | 78.1 |

more likely to fall outside the natural speech distribution on which the model is trained in Stage I (Figure 2). For more analyses on the effect of active selection see Appendix A.8.

Finally, we ask how well our approach preserves the text capabilities of the original text-based LLM backbone compared to the baselines. Table 5[6] shows that, unlike other speech-adapted models, which exhibit substantial forgetting, SALAD maintains the closest performance to the original text LLM. This result highlights the effectiveness of the distillation objective in constraining the model to remain faithful to its teacher while learning to achieve cross-modal alignment.

---

[5]https://huggingface.co/BAAI/bge-large-en-v1.5

[6]We use the DiVA model available at https://huggingface.co/WillHeld/DiVA-llama-3-v0-8b . While Held et al. (2025) report freezing a Llama-3-8B backbone, the released version on HuggingFace appears to be based on Llama-3.1-8B. Moreover, we found the released weights to differ from the Llama-3.1-8B checkpoint, which explains the differences in text performance reported in Table 5.

Table 5: SALAD best preserves the original text capabilities of the text-based LLM backbone after speech training compared to the baselines. "Acc." denotes the accuracy of the speech-adapted LLM given text input; "Gap" denotes the difference between the text-based LLM given text input and the "Acc." column. Gap cells are color-coded from best (green) to worst (red) for each task, where lower is better.

| | StoryCloze | | MMSU | | OBQA | | HellaSwag | | ARC-C | | PIQA | | Avg. | |
|---|---|---|---|---|---|---|---|---|---|---|---|---|---|---|
| | Acc. | Gap | Acc. | Gap | Acc. | Gap | Acc. | Gap | Acc. | Gap | Acc. | Gap | Acc. | Gap |
| Qwen2-Audio-7B | 81.1 | -0.2 | 46.9 | 1.3 | 73.4 | 3.3 | 69.4 | 2.6 | 68.6 | 3.4 | 76.0 | 2.8 | 69.2 | 2.2 |
| DiVA-Llama3.1-8B | 80.9 | 7.4 | 60.9 | 1.0 | 82.0 | 1.3 | 67.8 | 7.7 | 81.7 | 0.2 | **80.8** | 0.6 | 75.7 | 3.0 |
| GLM-4-Voice-9B | 81.6 | 17.2 | 48.4 | 17.8 | 69.9 | 17.8 | 70.6 | 9.9 | 70.9 | 22.4 | 77.9 | -3.4 | 69.9 | 13.6 |
| Qwen2.5-Omni-7B | 80.5 | 4.5 | 66.3 | 4.5 | 87.3 | 1.7 | 69.8 | 6.9 | 87.6 | 0.8 | 78.8 | 1.1 | 78.4 | 3.3 |
| SALAD-3B | 82.7 | 0.2 | 62.5 | -0.6 | 83.7 | -1.9 | 70.5 | 0.5 | 83.1 | **-1.2** | 78.6 | 0.0 | 76.9 | -0.5 |
| SALAD-7B | **84.9** | **-2.0** | **71.6** | **-0.8** | **90.1** | **-1.1** | **76.9** | 0.2 | **89.2** | -0.8 | 80.2 | **-0.3** | **82.2** | **-0.9** |

## 5  RELATED WORK

**Cross-modal transfer in speech-adapted LLMs.**   A large body of work has focused on transferring the general language capabilities of text LLMs to the speech domain. Deng et al. (2025); Wang et al. (2023); Tang et al. (2024); Chu et al. (2023; 2024); Hu et al. (2024) adapted text LLMs to process speech through multi-task supervised learning, combining tasks like speech recognition with speech instruction-following data to promote general-domain transfer. Other works, such as Fathullah et al. (2024); Wang et al. (2024); Held et al. (2025), addressed the scarcity of broad-domain supervised data by deriving rich supervision from unsupervised paired text–speech datasets via cross-modal distillation. Related to our method with $\alpha = 1.0$ in Equation 4, Wang et al. (2024); Held et al. (2025) showed that dense supervision—matching the full output distribution—in cross-modal distillation can outperform prior supervised-learning-based approaches. Several works (Wang et al., 2021; Held et al., 2025; Tseng et al., 2025) further promoted transfer through specialized representation alignment modules, typically transformer encoders with downsampling and query-based cross-attention, and Wang et al. (2024); Tseng et al. (2025) additionally matched sampling rates across modalities. While these works demonstrated varying degrees of cross-modal knowledge transfer, none matched the performance of text models of equivalent capacity on reasoning- and knowledge-intensive LLM benchmarks. Among open-training-recipe models, SALAD reduces the text–speech understanding gap by 11.7% over the next-best model and rivals the strongest closed-recipe system—within 1.2%—while using over an order of magnitude less training data and maintaining expressive, streaming-friendly speech representations. Our analysis of forgetting and misalignment also provides actionable insights into the factors driving the text–speech gap and the training dynamics of such models.

**Domain matching text–speech pretraining.**   Cuervo & Marxer (2024) demonstrated the benefits for speech LM training of using a TTS-synthesized spoken version of a small pretraining corpus designed to promote essential language capabilities. Zeng et al. (2025) synthesized an LLM pretraining corpus (FineWeb-Edu, corresponding to $\mathbb{D}_{\text{web}}$ in our study) for interleaved text–speech language modeling. Our active selection approach enables the use of far less synthetic speech by employing it adaptively to cover domain gaps in natural speech corpora. Moreover, we show that domain matching should be paired with cross-modal distillation to achieve general-domain alignment.

## 6  CONCLUSION

In this work, we studied the text–speech understanding gap, identifying forgetting of text capabilities and cross-modal misalignment as the two factors behind the underperformance of speech-adapted LLMs compared to their text-based counterparts. Building on this analysis, we introduced SALAD, a sample-efficient approach that combines cross-modal distillation with active data selection to target these challenges directly. Our experiments on 3B and 7B models demonstrated that SALAD achieves competitive performance with strong open-weight baselines, while using over an order of magnitude less data. These results suggest that carefully designed objectives and data selection strategies can substantially reduce reliance on costly, massive synthetic data or proprietary speech resources, paving the way for data-efficient methods to close the text–speech understanding gap.

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

# A APPENDIX

## A.1 MODEL DESCRIPTION

**Speech encoder.** The speech encoder transforms $l_a$ speech audio frames $\boldsymbol{a} \in \mathbb{R}^{l_a}$ into a sequence of $d_s$-dimensional latent speech representations $\boldsymbol{Z} \in \mathbb{R}^{l_s \times d_s}$ of length $l_s < l_a$. We use the Mimi speech tokenizer (Défossez et al., 2024), a causal lightweight speech encoder. Mimi produces multi-codebook representations $\boldsymbol{Z} \in \mathbb{R}^{l_s \times q \times d_s}$, where $q$ is the number of codebooks. We add the representations across codebooks, resulting in $\boldsymbol{Z} = \sum_{i=1}^{q} \boldsymbol{Z}_{:,i,:} \in \mathbb{R}^{l_s \times d_s}$. The speech encoder remains frozen.

**Adapter.** $\boldsymbol{Z}$ encodes low-level phonetic and acoustic information. The role of the adapter is to transform $\boldsymbol{Z}$ into $\boldsymbol{Z}' \in \mathbb{R}^{l_s \times d}$, a higher-level, more text-like representation. We implement the adapter as a stack of decoder-only transformer layers, preserving causality and streamability. If $d_s \neq d$, a linear projection is applied at the output to obtain $d$-dimensional representations. We evaluated several adapter sizes and found performance to saturate at around 122M parameters; this configuration was therefore adopted for all reported experiments. The adapter consists of 12 Llama-style decoder layers with residual dimension 960, MLP dimension 2560, and 15 attention heads with 5 KV heads.

**Language model.**   The language model processes multimodal sequences of text and speech representations $\boldsymbol{H} \in \mathbb{R}^{k \times d}$ and outputs a probability distribution over a vocabulary $\mathbb{V}_t$ of text tokens. Subsequences of $\boldsymbol{H}$ may correspond either to adapter-output speech sequences $\boldsymbol{Z}'$ or to sequences of text embeddings $\boldsymbol{E} \in \mathbb{R}^{l_w \times d}$ obtained by applying an embedding function to text tokens $\boldsymbol{w} = (w_1, \ldots, w_{l_w})$, with $w_i \in \mathbb{V}_t$. $\boldsymbol{H}$ is processed by a stack of transformer decoder layers, and the output logits at position $i$ are linearly predicted from the corresponding hidden representation, yielding a $|\mathbb{V}_t|$-dimensional logit vector that is Softmax-normalized into the probability distribution $P(w_{i+1} \mid \boldsymbol{H}_{\leq i})$. We initialize the language model from pretrained text language models from the Qwen2.5 family of LLMs (Qwen et al., 2025).

## A.2   TRAINING DETAILS: ANALYZING THE TEXT-SPEECH GAP

The models are trained with the AdamW optimizer (Loshchilov & Hutter, 2019) with a weight decay of $0.1$. We adopt a warmup-stable-decay learning-rate schedule (Hägele et al., 2024), consisting of a linear warmup of $500$ steps followed by a linear decay to zero over the final $20\%$ of training. The peak learning rate is tuned separately for each model size, with distinct values for the language-model backbone and the adapter. We use a learning rate of $10^{-3}$ for the adapter and $5 \times 10^{-5}$ for the LLM. All models are trained with a batch size of approximately 1M tokens and a context window of 2048 tokens.

## A.3   TRAINING DETAILS: CLOSING THE TEXT-SPEECH GAP

All models follow the hyperparameters and setup in Appendix A.2, except that SALAD-3B uses a learning rate of $10^{-3}$ for the adapter and $5 \times 10^{-5}$ for the LLM; SALAD-7B uses $10^{-4}$ for the adapter and $5 \times 10^{-6}$ for the LLM.

In Stage I, each batch is sampled with probability $2/3$ from $\mathbb{D}_{\text{speech}}$ and $1/3$ from the SmolLM corpus, following the common practice of mixing in pretraining data to mitigate forgetting (Béthune et al., 2025; Nguyen et al., 2024; Zeng et al., 2025). In Stage II, batches are drawn with equal probability from $\mathbb{D}_{\text{speech}}$, $\mathbb{D}_{\text{active}}$, and the SmolLM corpus (Allal et al., 2025).

For Stage II training of SALAD models, we resume from the last checkpoint before learning-rate decay and continue for 1.9B tokens with a linear decay of the learning rate. An exception is the experiments in Figure 8 and Table 8 (Appendix A.8), where training was resumed from the checkpoint after decay and continued for 950M tokens with a constant learning rate fixed to the post-decay value. These runs were conducted before we identified the setup that yielded stronger results for SALAD models.

During initial Stage II experiments, we observed that clusters with high misalignment often corresponded to domains such as academic citations (including DOIs and URLs) or non-English content, which produced hard-to-parse TTS artifacts. These clusters were inherently difficult to align not necessarily because of domain gaps, but because of mismatches between the TTS-generated speech and the ground-truth text. Consequently, we filtered out the top 15% of clusters (19 clusters for our 128-cluster setup) with the highest character error rate (CER), measured by transcribing the generated speech with Whisper-v3-large and comparing it to the ground-truth text. After removing these TTS-adverse clusters, we observed no meaningful correlation between CER and misalignment, and therefore attribute the remaining misalignment to domain gaps.

## A.4   CLUSTER ANNOTATION PIPELINE

We generate the domain annotations in Figures 2 and 9 by annotating embedding clusters with an LLM (Claude 3.7 Sonnet). Below, we outline the annotation and validation procedures. For a clustering model with $64$ clusters, the judge validated the annotations with an accuracy of $56\%$ (random chance is $1.6\%$).

**Phase A: Explain.**   For each cluster, we sample $k$ positives (closest to the centroid) and $n$ negatives (from the $M$ nearest neighbor clusters). An LLM is prompted with these examples to propose a short title, a descriptive label, and inclusion/exclusion criteria.

Listing 1: Explain prompt.

```
System:
You are an expert at explaining clusters of short texts.
Return STRICT JSON with fields:
  short_title, label, inclusion_criteria, exclusion_criteria, confidence

User:
{
  "cluster_id": int,
  "distance_metric": "cosine" | "l2",
  "positives": [ {"id": int, "distance": float, "text": str}, ... ],
  "negatives": [ {"id": int, "from_cluster": int,
              "distance": float, "text": str}, ... ],
  "instructions": "Name and describe ONLY the positive cluster concept.
              Use distances as prototypicality cues."
}
```

Listing 2: Judge prompt.

```
User:
{
  "categories": [ {"id": int, "title": str, "description": str}, ... ],
  "sample": { "text": str },
  "instructions": "Pick exactly one best-matching category id."
}
```

**Phase B: Judge.** We construct a catalog of all cluster labels and ask the LLM to classify holdout texts into exactly one cluster. This produces multiclass accuracy estimates per cluster and overall. The pipeline supports resumability and incremental judging, enabling efficient large-scale evaluation.

**Prompt format.** The LLM receives a JSON payload and is instructed to return JSON only. Listing 1 shows the structure of the EXPLAIN prompt.

The JUDGE prompt follows the same format, with a catalog of labeled clusters and a single sample to classify. Listing 2 shows its structure.

## A.5 EVALUATION PROTOCOL

We use StoryCloze (Hassid et al., 2023), MMSU and OpenBookQA from VoiceBench (Chen et al., 2024), HellaSwag (Zellers et al., 2019), ARC-Challenge (Clark et al., 2018), and PIQA (Bisk et al., 2020) for evaluation. In all cases, we synthesize spoken versions from the corresponding text data using the Kokoro TTS tokenizer with the `af-heart` speaker—including for benchmarks such as StoryCloze and VoiceBench—so as to maintain control over prompt formats, as described below. All are multiple-choice benchmarks. For each task, the model estimates the normalized log probability of each answer given the context, and accuracy is computed by checking whether the highest-scoring option matches the gold answer.

Table 6 presents the prompt templates used for each task. In the VoiceBench versions of MMSU and OpenBookQA, the model receives the list of answer options as part of the prompt context. However, we observed that smaller models, as well as models from the literature (e.g., DiVA), performed close to random under this setup. This behavior is consistent with prior findings that small language models often struggle with multi-choice formats when answer options are embedded in the input prompt (Allal et al., 2025). To address this limitation, we synthesized our own versions of MMSU and OpenBookQA with controlled prompt formats, including a continuation variant in which the model predicts the correct answer directly. For each model, we report performance under the prompt

format that yields the best results. We also observed differences between predicting the answer as the full option or just the letter label of that option. We also evaluate both variants.

Since most of our baselines are instruction-tuned models, we adopt when needed each model's native chat template: the "`Answer:`" segment is assigned to the assistant role, while the remaining context is presented as user input. To mitigate performance differences due to prompting, we further condition models on the task format using few-shot demonstrations. For each task, we include as many demonstrations as can fit in the audio context window of $2048$ tokens, up to a maximum of five. The number of shots used per task is: StoryCloze (5), MMSU (4), OpenBookQA (5), HellaSwag (3), ARC-Challenge (1), and PIQA (5). For each model, we report results with the best-performing prompt format.

Table 6: Evaluation task prompts. Placeholders are shown in `monospace`. Blue highlights denote text/audio inputs, and orange highlights denote continuations where likelihood is evaluated.

---

**StoryCloze**

`<story prefix>` `<answer>`

---

**MMSU (continuation)**

The following are multiple choice questions (with answers) about `<topic>` .

Question: `<question>`

Answer: `<answer>`

---

**MMSU (multiple choice)**

The following are multiple choice questions (with answers) about `<topic>` .

Question: `<question>`

A. `<A>`

B. ``

C. `<C>`

D. `<D>`

Answer: `<answer>`

---

**OpenBookQA (continuation)**

Question: `<question>`

Answer: `<answer>`

---

**OpenBookQA (multiple choice)**

Question: `<question>`

A. `<A>`

B. ``

C. `<C>`

D. `<D>`

Answer: `<answer>`

---

**HellaSwag**

`<context>` `<answer>`

---

**ARC-Challenge (continuation)**

Question: `<question>`

Answer: `<answer>`

---

**ARC-Challenge (multiple choice)**

Question: `<question>`

A. `<A>`

B. ``

C. `<C>`

D. `<D>`

Answer: `<answer>`

---

**PIQA**

`<question>`

`<answer>`

---

Table 7: Type-II Analysis of Covariance (ANCOVA) results controlling for $\alpha$ (fixed effects) and log(tokens). Left: dependent variable is average speech performance (%). Right: dependent variable is average text performance (%). Reported are degrees of freedom (df), sum of squares (SS), mean squares (MS), F-statistics ($F$), and corresponding $p$-values ($p$). Partial $R^2$ values quantify the unique variance explained by each factor.

| Speech (%) | | | | | Text (%) | | | | |
|---|---|---|---|---|---|---|---|---|---|
| Term | df | SS | MS | $F$ | $p$ | Term | df | SS | MS | $F$ | $p$ |
| log(misalignment) | 1 | 2.789 | 2.789 | 11.51 | 0.0025 | forgetting | 1 | 0.2588 | 0.2588 | 7.945 | 0.0097 |
| $\alpha$ (FE) | 4 | 2.255 | 0.5638 | 2.327 | 0.0867 | $\alpha$ (FE) | 4 | 0.4121 | 0.1030 | 3.163 | 0.0329 |
| log(tokens) | 1 | 38.39 | 38.39 | 158.4 | $8.5 \times 10^{-12}$ | log(tokens) | 1 | 0.0661 | 0.0661 | 2.031 | 0.168 |
| Residual | 23 | 5.573 | 0.2423 | – | – | Residual | 23 | 0.749 | 0.0326 | – | – |

Partial $R^2$: log(misalignment) $= 0.33$, $\alpha = 0.29$, log(tokens) $= 0.87$    Partial $R^2$: forgetting$= 0.26$, $\alpha = 0.35$, log(tokens) $= 0.08$

### A.6 Ordinary Least Squares Analysis

We fit ordinary least-squares models with (i) average speech performance (%) as

$$y_{\text{speech}} = \beta_0 + \beta_1 \log(\text{misalignment}) + \gamma_\alpha + \beta_T \log(\text{tokens}) + \varepsilon,$$

and (ii) average text performance (%) as

$$y_{\text{text}} = \beta_0 + \beta_1 \text{forgetting} + \gamma_\alpha + \beta_T \log(\text{tokens}) + \varepsilon,$$

where $\gamma_\alpha$ are fixed effects for $\alpha \in \{0, 0.25, 0.5, 0.75, 1\}$.

We report Type-II ANCOVA in Table 7: for each term, we compare the full model against the reduced model with that term removed (while keeping the other terms), yielding $SS_{\text{term}}$, $F$, and $p$. We also report partial $R^2 = SS_{\text{term}}/(SS_{\text{term}} + SS_{\text{resid}})$.

Both analyses indicate that, after controlling for $\alpha$ and training budget, misalignment is strongly associated with speech performance, and forgetting is strongly associated with text performance. To quantify out-of-sample explanatory power of the focal predictor beyond controls, we also compute the partial LOOCV $R^2$: $R^2_{\text{cv,full}}$ versus $R^2_{\text{cv,controls}}$. We obtain $\approx 0.34$ for misalignment (speech) and $\approx 0.23$ for forgetting (text), consistent with the ANCOVA results.

### A.7 Effect of Distillation on Text Performance

Figure 5 shows how the average text performance varies with the choice of $\alpha$ in Equation 4, where $\alpha \in \{0, 0.25, 0.5, 0.75, 1\}$. The models are trained on data drawn from $\mathbb{D} \in \{\text{Emilia+LibriHeavy, FineWeb-Edu}\}$ with training budgets ranging from 2B to 12B tokens. Overall, average text performance is less sensitive to these changes compared to average speech performance. Nonetheless, we observe a consistent, but small, improvement when training with the distillation objective ($\alpha = 1$) relative to the NLL objective ($\alpha = 0$).

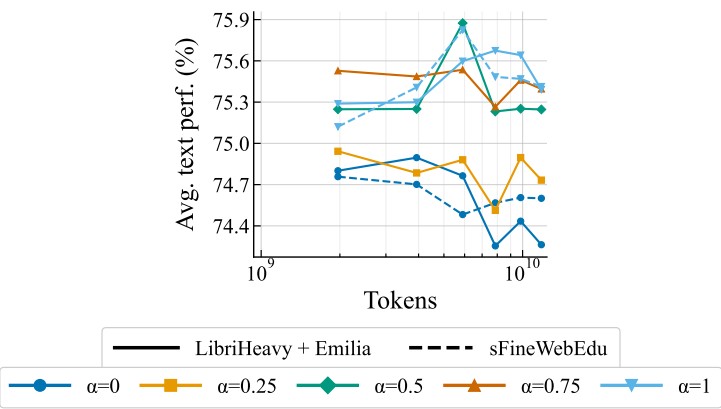

Figure 5: Impact of the training objective (controlled by $\alpha$ in Equation 4), the number of training tokens (shown on the $x$-axis), and the dataset choice on the average text performance.

## A.8 ACTIVE SELECTION ANALYSES

SALAD makes use of an active selection algorithm in Stage II to identify and cover gaps between natural speech and broad-domain text datasets. We analyze the role of this stage and study the impact of its hyperparameters on the overall performance. For these experiments, we apply Stage II training to the model trained with $\alpha = 1.0$ in Section 3. During Stage II, the model is trained for additional 950M tokens.

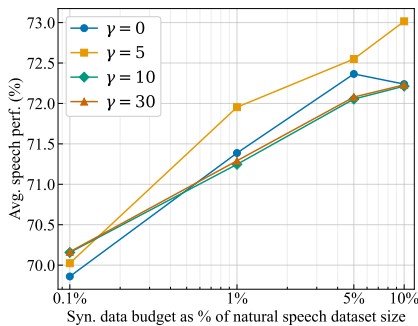

Figure 6: Stage II performance for SALAD-3B with different synthetic data budgets and selectivity $\gamma$ values ($K = 128$ clusters).

**Does two-stage training scale better than just first-stage training?** An important question is whether the improvements from the active learning stage stem from its design, or merely from the extra training steps added after Stage I. Figure 8 shows that Stage II yields consistent gains across tasks over Stage I for a given budget, with the exception of StoryCloze, where performance deteriorates slightly.

**Is active selection sensitive to the number of clusters used?** We present in Table 8 the performance for different values of the number of clusters $K \in \{64, 128, 256\}$, using $\gamma = 5$ and a Stage II synthetic data budget of 5% of the natural speech dataset size (LibriHeavy + Emilia). We find that varying $K$ has only a negligible effect on the overall average performance.

**How important is targeting domain gaps as more synthetic data is allowed?** While Table 3 and Figure 8 show meaningful Stage II gains—and Table 4 corroborates them—the role of domain-gap targeting remains unclear. Is it necessary to explicitly target domain gaps, or would tuning on a small amount of target, general-domain data suffice to improve alignment and performance? How does this trade-off evolve as we increase the selected data?

Table 8: Performance (Accuracy, %) of the SALAD-3B model after Stage II training with active selection (synthetic data budget of 5% of the natural speech data budget, $\gamma = 5.0$) across different number of clusters $K \in \{64, 128, 256\}$.

| $K$ | SC | MMSU | OBQA | HellaSwag | ARC-C | PIQA | Avg |
|---|---|---|---|---|---|---|---|
| 64 | **74.8** | 52.4 | 74.3 | **68.9** | **78.4** | 78.1 | 71.1 |
| 128 | 74.5 | **52.5** | **75.8** | 68.7 | **78.4** | 78.2 | **71.4** |
| 256 | 74.1 | 51.5 | 72.1 | 68.6 | 77.8 | **78.5** | 70.4 |

To investigate these questions, we vary $\gamma$ in Equation 8, which controls selectivity, and sweep the Stage II synthetic budget from 0.1% to 10% of the natural-speech dataset size (LibriHeavy + Emilia). We evaluate $\gamma \in \{0, 5, 10, 30\}$, where $\gamma = 0$ corresponds to uniform sampling from the target domain. Figure 6 reports average performance across spoken tasks as a function of budget. Active selection with $\gamma = 5$ consistently outperforms all other settings across budgets, *underscoring the importance of targeting domain gaps*. By contrast, over-focusing on gaps ($\gamma > 5$) yields gains only at very small budgets (0.1%) and falls behind even uniform sampling at larger budgets, suggesting that while selective targeting is beneficial, *maintaining exploration and diversity is equally crucial*.

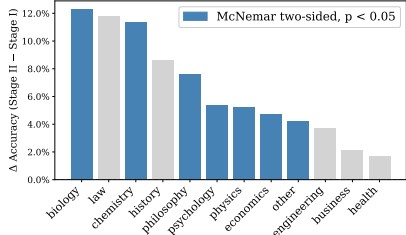

Figure 7: Improvements in accuracy between Stage I and Stage II for SALAD-3B across MMSU categories. We highlight the categories with statistically significant improvements.

**Does the active learning stage identify meaningful domain gaps?** Figure 7 breaks down the accuracy improvements across MMSU categories from Stage I to Stage II, showing that the largest statistically significant gains occur in categories such as biology and chemistry. Figure 9 shows the density of samples per cluster for the top-10 clusters selected by the active learning algorithm. Using the LLM-assisted annotation procedure described in Appendix A.4, we assign a domain label to each cluster and find that the most heavily sampled domain is *Molecular Biology*. These findings support our interpretation that Stage II boosts performance on these benchmarks by targeting more

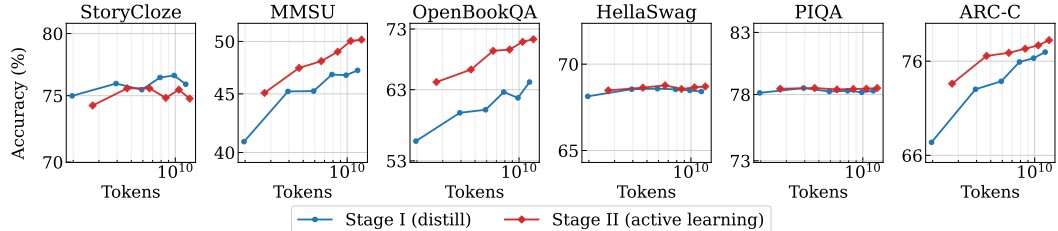

Figure 8: Impact of the two-stage training process on SALAD-3B, shown as a function of the number of training tokens and reported across spoken language understanding tasks.

Table 9: Impact of mismatched Stage II and evaluation speakers for SALAD-3B on VoiceBench MMSU and OpenBookQA.

| Evaluation speaker | MMSU | | OBQA | |
|---|---|---|---|---|
| | Acc. | Gap (%) | Acc. | Gap (%) |
| Kokoro TTS; `af-heart` voice (Original) | 52.5 | 9.4 | **76.7** | **5.1** |
| Kokoro TTS; `af-bella` voice | **53.0** | **8.9** | 76.0 | 5.8 |
| VoiceBench Google TTS; male voice | 52.4 | 9.5 | 74.3 | 7.5 |

technical domains. Moreover, they indicate that *our active learning algorithm effectively identifies and addresses meaningful domain gaps*, leading to measurable performance improvements.

**Do the gains of Stage II generalize across speakers?** Both our synthetic data from Stage II and the evaluations are generated with the same speaker, selected as the highest-quality voice available in our text-to-speech system. A natural concern is that the observed gains from Stage II might not generalize to other speakers. To address this concern, we evaluated on the original VoiceBench samples from MMSU and Open-BookQA—the tasks showing the largest improvements in Stage II—which use a different synthesizer and a speaker of the opposite gender from the one used in our Stage II training. Table 9 reports the results. While performance decreases slightly, a large fraction of the Stage II gain is preserved, indicating that *the improvements are robust and not tied to a single speaker's characteristics or synthesizers*.

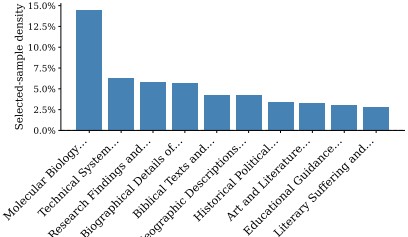

Figure 9: Distribution of Stage II-selected samples across top-10 most sampled domains.

### A.9    OPEN-ENDED GENERATION EVALUATIONS

SALAD is a pre-training method, and the standard way to evaluate *pre-trained* LLMs is through multiple-choice benchmarks, where likelihood-based accuracies can be computed—such as the benchmarks used throughout the main body of this paper. These evaluations directly align with the pre-training objective and are therefore well suited for base models. In contrast, the open-ended generation benchmarks commonly used for *instruction-tuned* LLMs rely on controlled, instruction-following behavior, and are not applicable to our base model, which does not reliably follow formatting or prompting constraints.

Nevertheless, to obtain a coarse sense of the generation quality of SALAD-trained models, we follow prior work on evaluating *pre-trained* LMs (Lakhotia et al., 2021; Maimon et al., 2025a; Eldan & Li, 2023) and compute three generation-focused metrics on samples generated from 300 LibriHeavy `test-clean` prefixes. For all generations, we use top-$k$ sampling with $k = 250$ and temperature 0.7.

Table 10: Generation-quality evaluation for Qwen2.5–3B and SALAD–3B. Metrics include GenPPL, AutoBLEU, and LLM-as-a-judge scores (1–10 scale). For GenPPL and AutoBLEU we report the median (Q1–Q3), and Holm–Bonferroni adjusted Mann–Whitney $p$-values against the Qwen2.5–3B baseline. Speech prompts for SALAD–3B use natural LibriHeavy speech or synthetic speech generated with Kokoro TTS `af-heart`.

| | GenPPL | | AutoBLEU | | LLM-as-a-judge | |
|---|---|---|---|---|---|---|
| | Median (Q1–Q3) | $p$-value | Median (Q1–Q3) | $p$-value | Median (Q1–Q3) | $p$-value |
| Qwen2.5–3B (baseline) | 3.519 (1.608–11.540) | – | 0.257 (0.195–0.302) | – | 7.50 (6.75–8.25) | – |
| SALAD-3B | | | | | | |
| Given text | 3.105 (1.504–7.660) | 0.651 | 0.269 (0.205–0.339) | 0.126 | 7.50 (7.00–8.25) | 1.000 |
| Given natural speech | 4.103 (1.793–9.233) | 1.000 | 0.247 (0.190–0.323) | 1.000 | 7.50 (7.00–8.25) | 1.000 |
| Given synthetic speech | 3.855 (1.960–9.028) | 1.000 | 0.245 (0.205–0.321) | 1.000 | 7.50 (7.00–8.25) | 1.000 |

- **GenPPL** (Lakhotia et al., 2021; Maimon et al., 2025a): perplexity of generated text under an independent reference LM (SmolLM-1.7B), assessing fluency;

- **AutoBLEU** (Lakhotia et al., 2021): a self-repetition metric quantifying diversity and penalizing over-repetitive behavior typical of base models; and

- **LLM-as-a-judge** scores (Eldan & Li, 2023), evaluating coherence, relevance, creativity, and fluency on a 1–10 scale. The prompt is shown in Listing 3. We use Claude 3.7 Sonnet with greedy sampling as judge.

To determine whether SALAD training affects generation quality relative to the text-only backbone, we report these metrics for SALAD-3B and its Qwen2.5–3B backbone model. For SALAD–3B, we evaluate generations conditioned on (i) text prompts, (ii) natural speech prompts, and (iii) synthetic speech prompts produced using the same setup as in our main evaluations—Kokoro TTS with the `af-heart` voice. All results, together with Welch's unpaired two-sample $p$-values computed against the Qwen2.5–3B backbone, are shown in Table 10. Across prompting modalities no metric shows any statistically significant difference relative to the Qwen2.5–3B backbone (Holm-adjusted Mann–Whitney tests), and we conclude no clear degradation in generation quality.

Listing 3: LLM-as-a-judge prompt.

```
System:
You are an expert literary critic and linguist, specialising in the
    evaluation of AI-generated creative text.

You MUST return ONLY a valid JSON object with this exact schema:
{
 "coherence": <int 1-10>,
 "relevance": <int 1-10>,
 "creativity": <int 1-10>,
 "fluency": <int 1-10>,
 "coherence_just": "<brief explanation>",
 "relevance_just": "<brief explanation>",
 "creativity_just": "<brief explanation>",
 "fluency_just": "<brief explanation>",
 "summary": "<one paragraph synthesis>"
}
Do NOT wrap the JSON in markdown backticks. Do NOT add any extra keys.

User:
INSTRUCTIONS FOR EVALUATION SCOPE:

The completions come from a base LLM truncated at a fixed length given a
    small book snippet as prefix, therefore:

1. Identify the core completion portion of the generated completion. E.g.
     ignore any trailing incomplete sentences or clearly off-topic text
    product of the LLM effectively switching to a different document.
2. Your analysis must stop at the end of the last complete sentence of
    this core completion.
3. Ignore any trailing incomplete sentences, loops, meta-commentary or
    tokens like <|endoftext|>.

SPECIAL CONSIDERATIONS FOR THIS EVALUATION (included ONLY for audio_bin /
     tts_mimi modalities):
* Punctuation Leniency: The prompt arrived via audio; if the generation
    continues the last sentence of the prompt, do not penalise sentence
    fusion.
* Name Leniency: Accept minor variations in proper nouns if the character
    's role and gender stays consistent.

Story Prompt:
> {STEM_PROMPT}

Generated Completion:
> {GENERATION}

Please provide the JSON rubric now.
```

