# OpenReview forum: "Closing the Gap Between Text and Speech Understanding in LLMs"
_ICLR.cc/2026/Conference — ICLR 2026 Poster_

### Official Review · Reviewer_VbQm · 2025-10-26

**Soundness:** 2
**Presentation:** 3
**Contribution:** 2
**Rating:** 6
**Confidence:** 4

**Summary:**

This paper addresses the text-speech understanding gap—the performance drop when speech-adapted LLMs process spoken inputs versus text inputs to text-based LLMs.
The authors decompose this gap into two factors: (i) catastrophic forgetting of text capabilities during speech adaptation, and (ii) cross-modal misalignment between speech and text representations.
Based on this analysis, they propose SALAD (Sample-efficient Alignment with Learning through Active selection and cross-modal Distillation), which combines cross-modal knowledge distillation with active data selection to improve alignment while mitigating forgetting. Applied to Qwen2.5 3B and 7B models, SALAD achieves competitive performance with recent speech-adapted LLMs while reportedly using much less training data.

**Strengths:**

1. Well-motivated problem decomposition: The two-factor analysis separating forgetting from cross-modal misalignment is conceptually clear and provides a useful framework. The mathematical formalization using KL divergence (Equations 2-3) enables quantitative measurement of both factors.
2. Rigorous empirical analysis (Section 3): The systematic study of how different training objectives (α parameter) affect forgetting and misalignment is valuable. The scaling law analysis with fitted curves (Table 2) and cross-validation provides insights into training dynamics.
3. Clear presentation and organization: The paper is well-written with logical flow from problem analysis to solution. Mathematical formulations are precise and figures effectively communicate key results.

**Weaknesses:**

1. Missing validation of core motivation: The paper motivates end-to-end approaches over cascaded systems by citing ability to capture "paralinguistic richness essential for natural spoken interaction" (Introduction). Yet no experiments evaluate paralinguistic understanding (emotion, prosody, speaker characteristics). The distillation objective enforcing identical text-speech distributions may actually suppress these cues, contradicting the motivation. This is very important because if content is the only thing that you want to model, then you can simply do cascaded system and in fact ASR+LLM backbone (Qwen2.5) is still significantly better as shown in Table 3.
2. Limited experimental scope: Only multiple-choice QA tasks—no open-ended generation. Would be curious to see if findings still hold in open-ended generation.
3. Entirely synthetic evaluation undermines validity: All benchmarks evaluate TTS-generated speech rather than natural speech. This raises questions about whether improvements generalize to real speech with acoustic variability, accents, and spontaneous characteristics. Table 8's limited test on VoiceBench shows performance degradation with different TTS speakers, but more analysis is needed.
4. Active selection assumes high-misalignment clusters represent domain gaps, but they could equally represent intrinsically difficult content or TTS artifacts. Some discussion around this would be helpful.

**Questions:**

Check weaknesses.

---

> ### Author Response · Authors · 2025-11-21
> **Response to reviewer VbQm**
>
> We thank the reviewer for the feedback. Please find our response to your comments below.
>
> > **Missing validation of core motivation:** The paper motivates end-to-end approaches over cascaded systems by citing ability to capture "paralinguistic richness essential for natural spoken interaction" (...) Yet no experiments evaluate paralinguistic understanding (...) This is very important because if content is the only thing that you want to model, then you can simply do cascaded system (...).
>
> While we indeed view the long-term goal of end-to-end systems as offering improvements over cascaded systems in terms of paralinguistic and acoustic awareness, we believe that an important initial step toward that goal is to make them as semantically coherent as cascaded and text-based systems, as discussed in the Introduction (Lines 65–71). We believe that if end-to-end systems possessed enhanced paralinguistic awareness but significantly worse semantic coherence than cascaded systems—which was the case for all non–closed-training-recipes models according to Table 3 and the findings of Chen et al. (2024) and Cui et al. (2025)—it would be difficult to argue for their practical adoption. Therefore, **the core motivation of our work is first to bridge the large semantic coherence gap in end-to-end systems comparable to that of cascaded and text-based systems**.
>
> To the best of our knowledge, our work is the first fully described method to demonstrate performance comparable to a strong cascaded system on general-domain reasoning- and knowledge-intensive benchmarks typically used for LLM evaluation, while relying exclusively on publicly available datasets. Note that although models such as Qwen2.5-Omni (Xu et al., 2025) and Kimi-Audio (KimiTeam et al., 2025) report similar performance, the details of their training methods and training data have not been disclosed, and are reported to rely on over an order of magnitude more data. Therefore, we consider our work to be a meaningful step toward addressing the limitations of end-to-end systems, and hopefully a stepping stone toward the long-term goals of end-to-end models.
>
> > (...) The distillation objective enforcing identical text-speech distributions may actually suppress these (paralinguistic) cues (...)
>
> Cross-modal distillation encourages the speech-adapted LLM to produce text outputs that match those of the text-only LLM, making its predictions invariant to paralinguistic variations in the speech signal. However, the model is end-to-end trainable and could be tuned to retain this information with an additional loss that focuses on paralinguistics—e.g. training for end-to-end speech-to-speech modeling as in Xu
> et al. (2025). Exploring how cross-modal distillation affects paralinguistic modeling in the model’s representations, as well as exploring speech-to-speech models, are both promising directions for future work.
>
> > Limited experimental scope: Only multiple-choice QA tasks—no open-ended generation. Would be curious to see if findings still hold in open-ended generation.
>
> SALAD is a **pre-training method**. To evaluate pre-trained models, multiple-choice benchmarks---where likelihood-based accuracies can be computed---are standard; indeed, all benchmarks used to evaluate our backbone model rely on likelihood comparison (Table 4 in Qwen et al., 2025). In contrast, we are not aware of any existing suitable open-ended generation benchmark. Given that pre-trained LLMs are compared based on NLL loss, in addition to it, we propose to compare models via misalignment (Eq. 2), which additionally reflects the existing text-speech gap we care about (see Figure 4).
>
> To provide a quantitative signal regarding the quality of open-ended generation, we have added Appendix A.9, which reports three generation-quality metrics
> computed on samples produced by SALAD-3B when prompted with 300 LibriHeavy `test-clean` prefixes.
> Specifically, we evaluate:
>
> - _Perplexity_ of the generated samples, measured by an independent reference
>   language model (SmolLM-1.7B; Allal et al., 2025).
>
> - _AutoBLEU_ (Lakhotia et al., 2021), which measures self-repetition and thus diversity.
>
> - _LLM-as-a-judge_ scores from Claude 3.7 Sonnet, assessing coherence, relevance,
>   creativity, and fluency on a 1–10 scale (the full evaluation prompt is provided in Appendix A.9).

---

> > ### Author Response · Authors · 2025-11-21
> > **Continuation of response to reviewer VbQm**
> >
> > To determine whether SALAD training affects generation quality relative to the text-only backbone, we report these metrics for SALAD-3B and its Qwen2.5–3B backbone model. For SALAD–3B, we evaluate generations conditioned on (i) text prompts, (ii) natural speech prompts, and (iii) synthetic speech prompts produced using the same setup as in our main evaluations—Kokoro TTS with the `af-heart` voice. All results, together with Welch’s unpaired two-sample p-values computed against the Qwen2.5–3B backbone, are shown in Table 10. **Across prompting modalities, no metric shows any statistically significant difference relative to the Qwen2.5–3B backbone, therefore we conclude no clear degradation in generation quality.**
> >
> > > Entirely synthetic evaluation undermines validity (...) This raises questions about whether improvements generalize to real speech
> >
> > Unfortunately, we were unable to find any multiple-choice natural-speech benchmark suitable for evaluating pre-trained models. Collecting such datasets remains an open challenge for the community. However, to address this concern, and as noted in the previous response, we compare in Appendix A.9 the distributions of the GenPPL, AutoBLEU, and LLM-as-a-judge scores for generations from SALAD-3B given natural-speech prompts from LibriHeavy and their corresponding synthesized versions. **For all metrics, there is no statistically significant difference between the generations conditioned on natural versus synthetic speech, indicating no measurable bias toward synthetic speech.**
> >
> > > (...) Table 8's limited test on VoiceBench shows performance degradation with different TTS speakers, but more analysis is needed.
> >
> > We note that our evaluations now include testing with a different speaker from the same TTS system (`af-bella` voice), as well as a different speaker from a different TTS system (VoiceBench). Both results are shown in Table 9 in Appendix A.8 and exhibit only minor degradation (an improvement of 0.5\% on MMSU and a drop of 0.7\% on OpenBookQA for `af-bella`, and a drop of 0.1\% on MMSU and 2.3\% on OpenBookQA for VoiceBench).
> >
> > > Active selection assumes high-misalignment clusters represent domain gaps, but they could equally represent intrinsically difficult content or TTS artifacts. Some discussion around this would be helpful.
> >
> > Thank you for raising this  point. This was indeed a preprocessing detail we unintentionally omitted from our submission.
> > In preliminary experiments, we observed a strong correspondence between the top misaligned clusters and the clusters with the highest TTS character error rate (CER) (as computed using Whisper-v3-large). These clusters corresponded to domains such as code, academic citations, and foreign languages, which tended to produce hard-to-parse TTS artifacts and consequently yielded only marginal improvements from active selection. As a result, we filtered out the top 15\% (19 clusters in our 128-cluster system) with the highest CER—estimated on the small per-cluster sample $\mathbb{D}_{\text{probe}}$ (Section 4.1, Line 399)—prior to Stage II. After filtering, we observed no meaningful correlation between TTS CER and misalignment, and we therefore attribute the remaining misalignment to domain gaps. Appendix A.8 further supports the conclusion that Stage II primarily targets domain gaps. We have added this clarification to Appendix A.3.

---

> ### Author Response · Authors · 2025-11-26
> **Follow-up on Rebuttal Discussion**
>
> Thank you again for your detailed feedback. We’re following up to see if there are any further questions or points we can address.

---

### Official Review · Reviewer_TPWY · 2025-10-31

**Soundness:** 3
**Presentation:** 3
**Contribution:** 3
**Rating:** 6
**Confidence:** 4

**Summary:**

This paper addresses the persistent performance gap between large language models (LLMs) adapted for speech input and their original text-based counterparts. While LLMs excel at text-based language understanding, adapting them to process speech directly results in a notable drop in performance - which is termed as "text - speech understanding gap".  The key contributions of this paper include:
1. it analyzes the text–speech understanding gap in a quantifiable way and diagnoses 2 main factors that may cause this gap: forgetting and cross-modal misalignment.
2. it proposes a 2-stage training strategy called SALAD that first does cross-modal distillation on natural speech data to improve alignment and mitigate forgetting, followed by an active synthetic speech sample selection to address domain coverage issue.

Empirical results show that SALAD achieves competitive performance compared to strong open-source speech-adapted LLMs, with much less speech data.

**Strengths:**

Originality:
This paper formalize the common phenomenon in speech-adapted LLMs where the LLMs exhibits significant gap between text and speech understanding. The paper also proposes a quantifiable metric to measure this phenomenon and provides statistical measure for both forgetting and cross-modal misalignment.

Quality:
1. The paper presents a thorough empirical evaluation, benchmarking SALAD against a wide range of open-source speech-adapted LLMs.
2. The analysis appears to be rigorous, accompanied with clear quantification of the factors contributing to the "text - speech understanding gap".

Clarity:
The paper is well organized and easy to follow.

Significance:
1. This work attempts to address a challenge when training a speech-LLM: it requires massive amount of speech data to achieve on-par speech understanding capabilities which is often not accessible. SALAD exhibits comparable performance while being less data-hungry.
2. The insights into the root cause of the text-speech understanding gap could be influential for future research on speech-LLMs.

**Weaknesses:**

1. The experiments focus on English speech and text, which couldn't address the generalizability to other languages or domains. Similarly, the synthetic data only contains 1 voice, which raises question about how the model can adapt to speaker variance.
2. Besides the quantitative results, it would be beneficial to provide some qualitative analysis to intuitively demonstrate how SALAD training minimize the text speech gap.

**Questions:**

1. In Table 3, How to interpret that SALAD-3B often achieves better/smaller text-speech gap than SALAD-7B? Doesn't larger model require more training steps?
2. Could there any negative impacts on model's capabilities in understanding audio/paralinguistic inputs when adopting SALAD training?
3. In Table 4, does the fact that SALAD-7B gets boosted on text task after SALAD training indicate that the training sets employed is giving SALAD models advantage in these tasks? (i.e. in-domain vs out-of-domain?)

I am giving a rating of 6 but am open to reconsider when authors answer these questions.

---

> ### Author Response · Authors · 2025-11-20
> **Response to reviewer TPWY**
>
> We thank the reviewer for the feedback. Please find our response to your comments below.
>
> > The experiments focus on English speech and text, which couldn't address the generalizability to other languages or domains.
>
> Due to the availability of ready-to-use public datasets, established benchmarks, baselines, and fast, reliable TTS systems solely for English, all results in the paper are shown for English. The
> text–speech understanding gap is a general problem, and the first necessary step is resolving it at least for one language. We leave exploration of other languages as a future work, when comprehensive evaluation can be done in other languages too.
>
> Regarding the domains used: for training, we use different general-domain text pretraining corpora for alignment; for evaluation, we rely on broad-domain knowledge and reasoning benchmarks, e.g. MMSU spans 12 diverse domains, while ARC consists of grade-school science problems.
>
> > (...) the synthetic data only contains 1 voice, which raises question about how the model can adapt to speaker variance.
>
> In original submission, in Table 9 (Appendix A.8) we evaluate SALAD-3B on the VoiceBench versions of MMSU and OpenBookQA—the benchmarks with the largest gains--with speakers different (different TTS model and gender) from our main evaluation voice (`af-heart`) , allowing us to assess the model’s ability to generalize across speaker characteristics. There is only a minor performance drop (0.1\% on MMSU and 2.3\% on OpenBookQA).
>
> Further, we assess robustness to speaker variation by testing the model with the `af-bella` voice, another high-quality speaker from Kokoro TTS, and have added these results to Table 8. These results again show minor performance differences (an improvement of 0.5\% on MMSU and a drop of 0.7\% on OpenBookQA).
>
> > Besides the quantitative results, it would be beneficial to provide some qualitative analysis to intuitively demonstrate how SALAD training minimizes the text–speech gap.
>
> To have qualitative analysis for distillation loss, we initially looked into open-ended generation and compared outputs between models trained with distillation loss and without it.
> However, it was hard to make any conclusion from the samples themselves, thus we focused on quantitative analysis measuring misalignment (see Figure 4), similar to standard LLMs where NLL is compared.
>
> For active learning we **did a qualitative analysis** by looking into what are the top-10 most sampled categories by SALAD (see Figure 9) and showing that these categories were not covered originally by speech corpora (compare with Figure 2).
> Consequently, Figure 7 shows the improvements we obtain per MMSU category.
>
> > In Table 3, how should we interpret that SALAD-3B often achieves a smaller text–speech gap than SALAD-7B? Doesn't the larger model require more training steps?
>
>
> We believe this stems primarily from a lack of hyperparameter tuning. For the 3B model we had sufficient compute resources to perform small-scale hyperparameter tuning (e.g., learning rate), whereas for the 7B model we were only able to run a single configuration.
>
> Regarding whether larger models require more training steps, we note that this is not the case for text LLMs. At the scale of billions of training tokens, larger models achieve lower losses than smaller models under the same number of training steps (see, e.g., Figure 1 in Touvron et al., 2023).
>
> _Touvron et al., “LLaMA: Open and Efficient Foundation Language Models”. In arXiv. 2023._
>
> > Could there any negative impacts on model's capabilities in understanding audio/paralinguistic inputs when adopting SALAD training?
>
> The cross-modal distillation objective we adopt in SALAD training encourages the speech-adapted LLM to produce text outputs that match those of the text-only LLM (regardless of the paralinguistic content of the speech). However, the model is end-to-end trainable and could be tuned to retain this information with an additional loss that focuses on paralinguistics—e.g. training for end-to-end speech-to-speech modeling as in Xu
> et al. (2025).

---

> > ### Author Response · Authors · 2025-11-20
> > **Continuation of response to reviewer TPWY**
> >
> > > In Table 4, does the fact that SALAD-7B gets boosted on text task after SALAD training indicate that the training sets employed is giving SALAD models advantage in these tasks? (i.e. in-domain vs out-of-domain?)
> >
> > Two factors may contribute to these improvements: (i) our pure distillation loss ($\alpha$ = 1 in Equation 4) trains on soft teacher distributions, which is known to behave similarly to label smoothing and can improve generalization (Hinton et al., 2015); and (ii) multimodal text–speech training exposes the model to multiple “views” (speech and text) of similar underlying content, while enforcing consistent text behavior across them, which may encourage more robust internal representations and in turn better generalization (Tian et al., 2020; Gupta et al., 2025).
> >
> > _Hinton et al. "Distilling the Knowledge in a Neural Network". In NIPS 2014 Deep Learning Workshop. 2014._
> >
> > _Tian et al. "Contrastive Multiview Coding". In Computer Vision – ECCV 2020. 2020._
> >
> > _Gupta et al. "Better Together: Leveraging Unpaired Multimodal Data for Stronger Unimodal Models". In arXiv. 2025._

---

> ### Author Response · Authors · 2025-11-26
> **Follow-up on Rebuttal Discussion**
>
> Thank you again for your helpful and encouraging feedback. Just checking in to see if there are any additional questions we can address.

---

### Official Review · Reviewer_nz78 · 2025-11-01

**Soundness:** 3
**Presentation:** 3
**Contribution:** 3
**Rating:** 6
**Confidence:** 3

**Summary:**

This paper identifies and analyzes the "text-speech understanding gap" in speech-adapted Large Language Models (LLMs), attributing it to two quantifiable factors: forgetting of pre-trained text capabilities and cross-modal misalignment. To bridge this gap, the authors propose SALAD, a two-stage method comprising: (1) Cross-modal distillation from the text-based LLM teacher to the speech-adapted student model, and (2) Active selection of a minimal amount of synthetic speech data to target residual domain misalignment. Experiments on 3B and 7B models show that SALAD achieves competitive performance with state-of-the-art models while using significantly less speech data.

**Strengths:**

- Formally defining and quantifying the text-speech gap via forgetting and misalignment.
- The paper is well-written and easy to follow.
- The paper shows that high performance can be achieved with significantly less data.

**Weaknesses:**

- The paper does not meet the standard for a thorough related work section, failing to properly situate itself within the current literature and justify its novelty.
- The paper fails to discuss and contrast its approach with highly relevant work, such as BLSP-KD and TASTE.
- The paper lacks of an ablation study on the hyperparameters, such as K and γ.

**Questions:**

- Given that  BLSP-KD already demonstrated the power of cross-modal distillation for this problem, what is the marginal contribution of the active learning component?
- Please provide an ablation on the cluster count K and the exponent γ.
- The paper choses a "worst-case" encoder to make a strong claim. However, how does SALAD compare against state-of-the-art non-causal encoders with built-in alignment mechanisms (e.g., TASTE)?

---

> ### Author Response · Authors · 2025-11-20
> **Response to reviewer nz78**
>
> We thank the reviewer for the feedback. Please find our response to your comments below.
>
> > The paper does not meet the standard for a thorough related work section
>
> To better contextualize our contributions **we have added a dedicated Related Work section (line 500).**
> We note that our original submission did include discussion of related work, though not in a standalone section due to space constraints (see Introduction, lines 72–87; Section 2, lines 131–133; Section 3, lines 186–192 and 256–258).
>
> > The paper fails to discuss (...) highly relevant work, such as BLSP-KD and TASTE
>
> We have further elaborated on how our work relates to both BLSP-KD and TASTE in the new **Related Work** section (line 500). We also discuss the value of our contributions in relation to these works in response to your other questions below.
> Please note that we cite both BLSP-KD (Section 1, Line 76; Section 3, Line 192) and TASTE (Section 1, Line 75; Section 3.4, Line 258) in our original submission, and per the [ICLR 2026 reviewer guidelines](https://iclr.cc/Conferences/2026/ReviewerGuide) "authors are not required to compare to papers solely on arXiv".
>
> > The paper lacks of an ablation study on the hyperparameters, such as K and $\gamma$
>
>  **Our initial submission already included ablations of the $\gamma$ hyperparameter in Appendix A.8, as referenced in Section 4.2, Line 424.** The results show that active selection with an appropriately chosen $\gamma$ consistently outperforms uniform selection—even when increasing the amount of synthetic data—thus active selection remains beneficial even with larger data pools. Regarding the number of clusters $K$, we have added Table 8 to Appendix A.8 with our initial experiments using $K \in \{64, 128, 256\}$:  **varying $K$ has only a negligible effect on the overall average performance.**
>
> > BLSP-KD already demonstrated the power of cross-modal distillation for this problem
>
> BLSP-KD used cross-modal distillation loss **only** for speech translation and an **artificially** created QA task. **In contrary, we show that** 1) cross-modal distillation loss is well-suited for bridging the performance gap between text and speech LLMs on harder reasoning- and knowledge-intensive tasks commonly used for text LLMs (e.g., MMLU, HellaSwag, ARC-Challenge); 2) there is a scaling behavior of different optimization objectives (see Section 3.5); 3) cross-modal distillation loss achieves general-domain text-speech alignment even when trained on narrow-domain data while standard NLL-based training fails to do so (despite its common use in the literature), even at larger data scales, and when text and speech training domains are matched.
>
> > what is the marginal contribution of the active learning component?
>
> **Our key contribution with the active learning component is showing that domain coverage is an important source of the text–speech gap, and that it can be addressed with a small amount of synthetic data through e.g. active selection.** Particularly:
>
> - Table 3 shows that Stage II (active selection) improves the average performance across tasks relative to Stage I (only cross-modal distillation) by 3.5\% and 1.5\% for the 3B and 7B models, respectively, with notable gains on tasks requiring specialized knowledge, such as MMSU, OBQA, and ARC-C, where performance improves by 5.2\% and 2.2\%, 11.2\% and 5.4\%, and 4.3\% and 2.3\% for the 3B and 7B models, respectively.
>
> - Figure 8 shows that, for any compute (token) budget, Stage II training outperforms longer Stage I training for our 3B model, with the greatest gains once again on MMSU, OBQA, and ARC-C, and with roughly constant gains across compute budgets of approximately 3\%, 7\%, and 1\%. This indicates that the benefits of Stage II do not arise merely from longer cross-modal distillation training but—as discussed in Section 3.2—stem from matching speech and text over a broader domain.
>
> - Table 4 and Figure 6 further show that active selection enables higher behavioral alignment—as measured by performance on broad-domain downstream tasks—than uniformly sampling data from the general-domain distribution under the same data budget, with average gains of up to 1.6\% relative to uniform sampling for our 3B model.

---

> > ### Author Response · Authors · 2025-11-20
> > **Continuation of response to reviewer nz78**
> >
> > > (...) how does SALAD compare against state-of-the-art non-causal encoders with built-in alignment mechanisms (e.g., TASTE)?
> >
> > Among the baselines **we compare against, DiVA employs an explicit input–representation alignment objective similar to that of TASTE.** The other baselines also incorporate speech–text alignment, as their encoders are derived through temporal downsampling of Whisper encoder representations, which were trained as speech-to-text transducers.
> >
> > Given the superior or comparable performance of SALAD relative to these baselines, our results suggest that explicit input–representation alignment may not be necessary for achieving cross-modal alignment, provided that an appropriate training objective and sufficiently matched domains are used. **Nevertheless, we note that input–representation alignment is orthogonal to our method and could potentially complement it to further reduce the text–speech understanding gap.** Future work could explore the effect of different encoders on the text-speech understanding gap.

---

> ### Author Response · Authors · 2025-11-26
> **Follow-up on Rebuttal Discussion**
>
> Thank you again for your thoughtful feedback. We’re checking in to see if there are any further questions or clarifications we can address.

---

### Author Response · Authors · 2025-12-02

Dear Reviewers and Area Chair,

We thank the reviewers for their thoughtful and constructive feedback. We are encouraged that they highlighted the contribution of **“formaliz[ing] the common [text–speech understanding gap] phenomenon in speech-adapted LLMs”** and providing **“a useful framework”** for quantifying forgetting and cross-modal misalignment (Reviewers TPWY, VbQm), the **“thorough empirical evaluation”** and **“rigorous”** empirical analysis (Reviewers TPWY, VbQm), and that the paper is **“well-written and easy to follow”** and our method **“achieves comparable performance… with much less speech data”** (Reviewers nz78, TPWY). We sincerely appreciate their time and effort.

The main points of feedback raised were: (i) clearer positioning relative to related approaches; (ii) more complete ablations of the active selection hyperparameters $K$ and $\gamma$; (iii) robustness to speaker variation and reliance on synthetic speech; (iv) limited open-ended and paralinguistic analysis; and (v) assumptions behind the active selection strategy.

We have addressed all questions and revised the manuscript accordingly. Key updates include: a **dedicated Related Work section** (Line 500); expanded **ablation studies, including new \(K\) experiments** (Line 1045); **robustness evaluations with additional speakers and TTS systems** (Line 1105); **open-ended generation analyses** and natural vs synthetic prompt comparisons (Line 1121); and added clarification of the **active selection filtering step** (Line 791).

We also clarified the core motivation of our work: reducing the text–speech performance gap on language understanding tasks that currently separates end-to-end speech LLMs from strong cascaded and text-based systems. Our results show that SALAD makes progress toward this goal and, to our knowledge, is the first fully described method to reach cascaded-level performance on the general-domain reasoning- and knowledge-intensive benchmarks typically used for LLM evaluation, while relying only on publicly available datasets. While paralinguistic modeling is important for the long-term development of speech LLMs, it is orthogonal to this primary objective and better suited for future work.

Best regards,

Authors

---

### Meta-Review · Area_Chair_nH9t · 2025-12-19

**Summary:**

(*Disclaimer: given the peculiar review process, some of my choices and reasonings below will be highly subjective, as I tried to imagine how a reviewer would have reacted to a specific response. I understand that any negative choice will be perceived as unfair by the authors, and I apologize in advance for that.*)

(*Second disclaimer: the authors and some reviewers explicitly mention some changes in scores that occurred during the rebuttal. As these were reverted due to the possibility of collusion in light of the security incident, I will tend to disregard this information.*)

The paper introduces SALAD, a method for adapting pre-trained LLM models to work with audio signals. In particular, they distinguish between two separate issues that can arise when adding an end-to-end audio module (forgetting of text knowledge and cross-modality misalignment), and they design a two-step procedure to solve both. A wide set of experiments validate their proposal.

The paper only had three reviews, all suggesting a weak acceptance. The two main concerns were: (a) the paper claims end-to-end training can allow the LLM to work on paralinguistic properties of the audio signal, but their training is explicitly designed to remove these cues; and (b) the experimental evaluation can be improved (more languages, more speakers, natural-language speech, etc.).

The authors provided a strong rebuttal, with several new experiments over the appendices. However, no reviewer participated in the rebuttal, which is extremely sad.

My own evaluation below is that the rebuttal answered a good portion of the reviewers' concerns. Because the original evaluations were all positive, it is reasonable to think that at least one or two reviewers would have moved to a strong acceptance of the paper. Since all reviewers also agreed on the significance of the topic and the technical validity of the solution, I am recommending acceptance of the paper.

**Reviewer Concerns:**

(*I will focus on some key weaknesses identified by multiple reviewers.*)

**Paralinguistic clues** (`VbQm`, `TPWY`): the authors argue that their main motivation is closing the gap between an end-to-end system and a cascaded system, since models that work well on paralinguistic tasks but poorly on other tasks will not be used in practice. I find this a good motivation.

**More experiments** (`VbQm`, `TPWY`): `VbQm` asked for experiments on open-ended generation and natural speech; `TPWY` asked for datasets with more than one speaker and language. The authors partially answered several of these concerns via additional experiments, in the limits of what open datasets are available.

**Reviewer Scores:**

`VbQm`: most concerns were addressed in the rebuttal. It is probable the reviewer would have increase its evaluation to a strong acceptance.

`TPWY`: same as `VbQm`.

`nz78`: this is a bit tricky. The reviewer asked for a better related work section, which the authors addressed. However, they also asked for additional comparisons to arXiv papers (violating ICLR rules), and for some experiments that were already present in the paper (albeit in the appendix). Due to this, I mostly ignored these concerns in my evaluation.

---

### Decision · Program_Chairs · 2026-01-26

Accept (Poster)